# Mechanism Actions of Coniferyl Alcohol in Improving Cardiac Dysfunction in Renovascular Hypertension Studied by Experimental Verification and Network Pharmacology

**DOI:** 10.3390/ijms251810063

**Published:** 2024-09-19

**Authors:** Qiuling Wu, Qilong Zhou, Chengyu Wan, Guang Xin, Tao Wang, Yu Gao, Ting Liu, Xiuxian Yu, Boli Zhang, Wen Huang

**Affiliations:** 1Natural and Biomimetic Medicine Research Center, Tissue-Orientated Property of Chinese Medicine Key Laboratory of Sichuan Province, West China School of Medicine, West China Hospital, Sichuan University, Chengdu 610000, China; lingwq570@163.com (Q.W.); qlzhou@scu.edu.cn (Q.Z.); wanchengyu@wchscu.cn (C.W.); xinguang@scu.edu.cn (G.X.); terrywang1126@scu.edu.cn (T.W.); gym3377@163.com (Y.G.); liuting526@wchscu.cn (T.L.); yuxiuxian@wchscu.cn (X.Y.); 2Innovative Chinese Medicine Academician Workstation, West China Hospital, Sichuan University, Chengdu 610000, China

**Keywords:** coniferyl alcohol, renovascular hypertension, myocardial hypertrophy, network pharmacology, inflammatory response, MMP9/COX2/TNF α/IL-17

## Abstract

Renovascular hypertension (RH), a secondary hypertension, can significantly impact heart health, resulting in heart damage and dysfunction, thereby elevating the risk of cardiovascular diseases. Coniferol (CA), which has vascular relaxation properties, is expected to be able to treat hypertension-related diseases. However, its potential effects on cardiac function after RH remain unclear. In this study, in combination with network pharmacology, the antihypertensive and cardioprotective effects of CA in a two-kidney, one-clip (2K1C) mice model and its ability to mitigate angiotensin II (Ang II)-induced hypertrophy in H9C2 cells were investigated. The findings revealed that CA effectively reduced blood pressure, myocardial tissue damage, and inflammation after RH. The possible targets of CA for RH treatment were screened by network pharmacology. The interleukin-17 (IL-17) and tumor necrosis factor (TNF) signaling pathways were identified using a Kyoto Encyclopedia of Genes and Genomes (KEGG) enrichment analysis. The inflammatory response was identified using a Gene Ontology (GO) enrichment analysis. Western blot analysis confirmed that CA reduced the expression of IL-17, matrix metallopeptidase 9 (MMP9), cyclooxygenase 2 (COX2), and TNF α in heart tissues and the H9C2 cells. In summary, CA inhibited cardiac inflammation and fibrohypertrophy following RH. This effect was closely linked to the expression of MMP9/COX2/TNF α/IL-17. This study sheds light on the therapeutic potential of CA for treating RH-induced myocardial hypertrophy and provides insights into its underlying mechanisms, positioning CA as a promising candidate for future drug development.

## 1. Introduction

Hypertension, a prevalent and incurable chronic disease, poses a significant threat to cardiac health [1]. Renovascular hypertension (RH) stands out as a type of secondary hypertension, primarily induced by arterial stenosis, which often manifests as unilateral or bilateral renal artery stenosis [2]. This stenosis triggers an increase in angiotensin II (Ang II) production and aldosterone release, activating the renin–angiotensin system, elevating blood pressure, and potentially leading to cardiac insufficiency [3,4]. It is important to note that RH carries a higher risk of heart disease compared to essential hypertension [5]. Recent studies have also shown enhanced cardiac inflammation in animal models of RH, resulting in more severe cardiac remodeling, fibrosis, and compromised cardiac function [6,7]. Therefore, research on treatment for RH is of great significance for preventing and controlling the progression of the aforementioned diseases.

Coniferyl alcohol (CA) is widely present in traditional Chinese herbal remedies, specifically in Picea neoveitchii Mast and Fructus Aurantii [8]. CA frequently serves as a precursor for synthesizing chemicals like ferulic acid, vanillin, and silybin, all known for their vasodilatory and antihypertensive properties [9,10,11]. Additionally, CA itself exhibits vasodilatory effects [12,13]. However, the therapeutic efficacy of CA in treating hypertension, as well as its potential to ameliorate hypertension-induced cardiac dysfunction, remains unexplored and awaits further investigation.

Network pharmacology, through the systematic integration and interpretation of biological systems, facilitates a comprehensive understanding of disease progression and evolution [14]. With an emphasis on revealing the mechanisms of action of drugs through multi-pathway and multi-target signaling pathways, this methodology provides many opportunities for the discovery of innovative therapeutic agents [15]. Network pharmacology is an innovation way in the research pathway of “diseases–targets–drugs” in medicine. It integrates and summarizes information from existing databases through a series of methods to determine effective drug intervention treatments for diseases [5]. The multi-target pattern of network pharmacology provides a more reasonable explanation for diseases and complications, facilitating disease identification and drug combination therapy [16,17]. Network pharmacology has been widely applied in cardiovascular diseases, such as for identifying the effective components and potential pharmacological mechanisms of Danshen Decoction in the treatment of cardiovascular diseases [18], clarifying the effect of QiShenYiQi Dripping Pills (T101) in the treatment of heart failure depending on multi-components, multi-targets, and multi-pathways [19], and revealing the efficacy and potential mechanism of anxia Baizhu Tianma Decoction in the treatment of hypertension [20]. Additionally, the integration of network pharmacology with molecular docking, a technique for evaluating the binding affinity between receptors and drug molecules, has emerged as a pivotal tool in drug discovery and development [21]. This combined approach holds great potential, particularly in situations where the targets of drugs therapy are not yet clear [22]. 

In this study, by applying the aforementioned methodologies, the intricate relationship between CA and RH was uncovered and the binding affinity between CA and its related targets was studied in detail. The protective efficacy and underlying mechanisms of CA against cardiac hypertrophy induced by RH were illuminated through experimental verification. This research presents novel perspectives and methodologies for the treatment of RH.

## 2. Results

### 2.1. The Protective Effect of CA on Cardiovascular Parameters and Cardiac Hypertrophy in RH Mice

In this study, the targets and signaling pathways for the CA treatment of RH myocardial hypertrophy were identified using network pharmacology. A total of 111 CA targets were obtained from TargetNet (scores > 0.1) (Appendix A), 88 CA targets were obtained from SuperPred (Appendix A), 25 CA targets were obtained from HERB (Appendix A), and 70 CA targets were obtained from SEA (scores ≥ 0.4) (Appendix A). By summarizing the results of the above four databases and removing duplicates, 227 CA targets were obtained (Appendix A). In total, 437 RH targets were obtained from GeneCards (Appendix A), 778 RH targets (scores ≥ 10) from CTD (Appendix A), 72 RH targets were obtained from OMIM (Appendix A), and 78 RH targets were obtained from DisGeNET (Appendix A). The results of the four databases were summarized, and 1227 RH targets were obtained by removing duplicates (Appendix A). A total of 52 intersection targets were obtained by comparing the CA and RH targets (Figure 1), and these targets are detailed in Table 1. Thus, CA may be useful for the treatment of RH. Therefore, we established a representative RH animal model using 2K1C mice for verification. Four weeks after surgery, the systolic and diastolic pressures of the 2K1C mice were obviously higher than those in the sham group, demonstrating the successful establishment of the RH mouse model (Figure 2A,B). Systolic and diastolic pressures were significantly decreased after three weeks of continuous administration of 10 mg/kg BENA and 20 and 40 mg/kg CA (Figure 2A,B). Without affecting body weight, 10 mg/kg BENA and 40 mg/kg CA significantly reduced the heart weight/body weight (HW/BW) ratio (Figure 2C,D). Echocardiographic indices were monitored simultaneously. As shown in Figure 3, the IVS, LVPW, EF, and FS indices of the heart showed obvious hypertrophy in the model group. These changes were relieved to varying degrees after the administration of 10 mg/kg BENA and 40 mg/kg CA. H&E staining of the heart tissues showed obvious myocardial cell injury, inflammatory penetration, and necrosis in the model group, which could be alleviated by 10 mg/kg BENA and 40 mg/kg CA (Figure 4A,B). These results indicated that CA and BENA had significant inhibitory effect on RH-induced cardiac inflammation and hypertrophy. 

### 2.2. Network Pharmacology and Molecular Docking Analysis

The mechanisms of CA in reducing blood pressure and alleviating myocardial inflammation and hypertrophy after RH were predicted using network pharmacology and molecular docking. The intersection targets were imported into Cytoscape to establish a “drug–target–disease” network (Figure 5A). The information on core targets (degree > 20) is shown in Table 2. A GO analysis was used to identify the top ten molecular functions (MFs) (Figure 5B), biological processes (BPs) (Figure 5C), and cellular components (CCs) (Figure 5D). A KEGG analysis enriched the disease pathways, including the IL-17 and TNF signaling pathways (Figure 5E). The IL-17 and TNF signaling pathways play key roles in the regulation of disease inflammation [23,24]. Based on the above results showing that CA alleviated cardiac inflammation after RH, the KEGG results also indicated that the IL-17 and TNF signaling pathways were closely related to the core targets. Therefore, we conducted the molecular docking of CA with the core targets and IL-17. The specific docking mechanisms between CA and the core targets are shown in Figure 6. Those with a lower docking binding energy were MMP9 (−6.02 kcal/mol), TNF α (−6.01 kcal/mol), COX2 (−5.04 kcal/mol), and IL-17 (−5.07 kcal/mol), which were associated with inflammation. It was speculated that CA might affect myocardial inflammation and hypertrophy in RH models by regulating the expression of key proteins in the IL-17 and TNF signaling pathways.

### 2.3. CA Alleviated RH-Induced Myocardial Inflammation and Hypertrophy by Decreasing the Expression of MMP9/COX2/TNF α/IL-17

CA may attenuate cardiac inflammation and myocardial hypertrophy in the RH model through the TNF and IL-17 signaling pathways. Thus, we examined the protein expressions of the main targets of the inflammatory signaling pathways in cardiac tissues before and after CA treatment. The serum levels of TNF α and IL-17 were significantly increased in the model group compared to the sham group (Figure 7A,B). The expressions of TNF α, IL-17, COX2, and MMP9 in the heart tissues of the RH group were obviously enhanced by Western blot analysis (Figure 7C–F). The 40 mg/kg CA treatment significantly suppressed the expressions of these proteins (Figure 7C–F). These results suggest that CA regulated cardiac inflammation and hypertrophy in the RH mice by down-regulating MMP9/COX2/TNFα/IL-17, thereby inhibiting the TNF and IL-17 signaling pathways. 

### 2.4. CA Reduced Ang II-Induced Hypertrophy of H9C2 Cells by Decreasing the Expression of MMP9/COX2/TNF α/IL-17

H9C2 cells were treated with different concentrations of Ang II (0.04, 0.2, 1, 5, and 25 μM) for 48 h. Their cell viability decreased significantly at concentrations above 5 μM (Figure 8A). The exposure of the H9C2 cells to CA (0.5, 5, and 50 μM) for 24 h did not affect their cell viability (Figure 8B), but 50 μM CA significantly alleviated the decreased cell viability caused by 5 μM Ang II (Figure 8C). Further validation of the animal experimental results was achieved by stimulating the H9C2 cells with Ang II in vitro. This stimulation markedly up-regulated α-smooth muscle actin (α-SMA) expression, indicating successful hypertrophy induction by Ang II (Figure 9A). Consistent with the in vivo results, the expressions of TNF α, IL-17, COX2, and MMP9 were up-regulated in the H9C2 cells stimulated by Ang II at 5 μM and were significantly down-regulated by CA at 50 μM (Figure 9B–E).

### 2.5. Effects of CA Toxicity on Brain, Liver, Lungs, and Spleen of Mice

After administrating CA for three weeks, the brain, liver, lung, and spleen tissues were stained with H&E. As shown in Figure 10, there was no inflammatory infiltration or cell necrosis in the brain, liver, lungs, or spleen tissues of any group, suggesting that CA itself did not cause damage to other organs.

## 3. Discussion

In this study, based on reports of CA vasodilation [12], we demonstrated, for the first time, that CA had the effect of lowering blood pressure and alleviating cardiac inflammation in an RH model through the combination of a network pharmacological analysis and experimental testing. We found that CA ameliorated cardiac inflammation and myocardial hypertrophy after RH by regulating the expressions of IL-17, TNF α, MMP9, and COX2 in both the IL-17 and TNF signaling pathways. Moreover, similar verification was further obtained with H9C2 cells in vitro.

We conducted a comprehensive investigation into the antihypertensive and cardioprotective effects of CA within an RH model. Experimental data indicated that both CA and BENA (positive control) significantly reduced diastolic and systolic blood pressures in the RH model. It is worth noting that 10 mg/kg of BENA, as well as 20 and 40 mg/kg of CA, had no significant impact on the weight of the mice, while 10 mg/kg of BENA and 40 mg/kg of CA strikingly reduced the HW/BW ratio. Further echocardiographic analysis and H&E staining of heart tissues indicated that 10 mg/kg of BENA and 40 mg/kg of CA were effective in alleviating the cardiac inflammation and hypertrophy induced by RH. These results clearly indicate that CA can reduce RH-induced cardiac inflammation and hypertrophy while reducing blood pressure. Previous reports have indicated that BENA can reduce the incidence of cardiovascular events in the treatment of hypertension [25]. A similar conclusion was also obtained in this study.

To gain a deeper understanding of how CA exerts its antihypertensive and cardioprotective effects, we conducted a detailed investigation of its mechanisms using network pharmacology. The findings revealed that TNF, IL-17, COX2, and MMP9 were the primary genes involved in CA’s action against RH. Further KEGG analysis highlighted the critical roles of the TNF and IL-17 signaling pathways in this process. Molecular docking techniques also demonstrated an excellent binding affinity between CA and these core targets. Hypertension is often accompanied by systemic inflammation, triggering the excessive secretion of proinflammatory cytokines, which can lead to functional abnormalities such as cardiac hypertrophy [26]. Notably, when cells sense inflammatory stimuli, COX2 exhibits a high level of responsiveness to various pro-inflammatory mediators and cytokines [27]. Meanwhile, pro-inflammatory cells in the heart release MMP9, and its activation not only intensifies the inflammatory response, but also disrupts the balance of the extracellular matrix (ECM), ultimately impairing cardiac function [22]. In our study, we observed that 40 mg/kg of CA significantly down-regulated the expression levels of TNF α, IL-17, COX2, and MMP9 in heart tissues, but 10 mg/kg of BENA did not significantly reduce their expression. These results suggest that CA alleviated RH-induced cardiac inflammation and myocardial hypertrophy by reducing the expression of MMP9/COX2/TNF α/IL-17, and that the therapeutic mechanism of 10 mg/kg of BENA did not depend on them. These results strongly indicate that CA improves cardiac inflammation and myocardial hypertrophy in RH by regulating the expression of MMP9/COX2/TNF α/IL-17 in the TNF and IL-17 signaling pathways.

Previous studies on small-molecule drugs in diseases have shown that drugs often affect protein stability by binding to proteins and affecting their abundance, which are more common in inflammatory, tumor, and autoimmune diseases. It is common for drugs to regulate protein expression by interacting with proteins to induce ubiquitination, phosphorylation, and acetylation, or to reduce protein abundance to affect stability [28,29,30]. Therefore, CA, a small-molecule drug derived from traditional Chinese medicine, may have a similar pattern of action to other small molecules. In cardiac inflammation caused by renovascular hypertension, CA may interact with TNF α, IL-17, COX2, and MMP9 to affect their protein stability and reduce their expressions, thus playing a role in improving inflammation. The interaction between CA and TNF α, IL-17, COX2, and MMP9 needs to be further explored through pull-down, surface plasmon resonance analyses, and other experiments, which will be the primary problem to be solved in our subsequent research.

Excessive Ang II in RH has been proven to induce cardiac hypertrophy [31]. Current research indicates that Ang II is a central factor causing myocardial hypertrophy [32]. To further investigate this phenomenon, we successfully established an AngII-stimulated hypertrophy model of H9C2 cells in vitro [33]. The degree of cardiomyocyte hypertrophy was assessed by observing the expression of α-SMA [34]. The results showed that CA can effectively reduce the Ang II-induced hypertrophy of H9C2 cells and significantly down-regulate the expression of inflammation-related proteins such as TNF α, IL-17, COX2, and MMP9. These findings suggest that CA alleviates the hypertrophy of cardiomyocytes by reducing the expressions of these inflammatory proteins.

## 4. Materials and Methods

### 4.1. Reagents and Materials 

CA (CAS number: 458-35-5) was obtained from Adamas (Shanghai, China). A tumor necrosis factor α (TNF α) assay kit (SEKM-0034) was obtained from Solarbio (Beijing, China). An interleukin-17 (IL-17) kit (orb775012) was purchased from Biorbyt (Cambridge, UK). Hematoxylin and eosin (H&E) staining was performed using Baso Cell (Zhuhai, China). Antibodies against TNF α (AF7014) and IL-17 (DF6127) were obtained from Affinity Biosciences (Changzhou, China). Antibodies against cyclooxygenase 2 (COX2) (sc-514489) and matrix metallopeptidase 9 (MMP9) (sc-393859) were procured from Santa Cruz Biotechnology (Santa Cruz, TX, USA). The GAPDH antibody (60004-1-Ig), goat anti-rabbit IgG-HRP (SA00001-2), and goat anti-mouse IgG-HRP (SA00001-1) were bought from Proteintech (Wuhan, China).

### 4.2. Prediction of Targets of CA and RH

CA targets were obtained from TargetNet (http://targetnet.scbdd.com, accessed on 13 July 2023), SuperPred (https://prediction.charite.de/, accessed on 13 July 2023), a high-throughput experiment and reference-guided database (HERB) (http://herb.ac.cn, accessed on 13 July 2023), and a similarity ensemble approach (SEA) (https://sea.bkslab.org/, accessed on 13 July 2023) [35].

RH targets were acquired from GeneCards (https://www.genecards.org/, accessed on 10 July 2023), a comparative toxicogenomics database (CTD) (http://ctdbase.org/, accessed on 10 July 2023), online mendelian inheritance in man (OMIM) (https://www.omim.org/, accessed on 10 July 2023), and the disease gene network (DisGeNET) (https://www.disgenet.org/, accessed on 10 July 2023). 

### 4.3. Animals and Experimental Design

For animal experiments, 6–8-week-old male C57BL/6 mice were obtained from the SPF company (license number: SCXK jing 2019-0010). They were housed in a room with a 12 h light/12 h dark cycle (23 ± 2 °C), where food and water were freely available. The experimental design was approved by the Ethics Committee of West China Hospital, Sichuan University (Ethics record number: 20220302077).

After adaptation, the animals were divided into five groups: sham (saline), model (saline), 10 mg/kg benazepril (BENA), 20 mg/kg CA, and 40 mg/kg CA. CA and BENA were administered intragastrically once daily for three weeks.

### 4.4. 2K1C Hypertension Model

RH was induced in the mice using a two-kidney, one-clip (2K1C) operation, as previously described [36,37]. The mice were anesthetized with isoflurane. After disinfecting the skin on the back, the right kidney was exposed via an incision. The right renal artery was divided and silver clips were placed around it to constrict the blood flow. Finally, the incision was carefully sutured.

### 4.5. Blood Pressure Measurement

The mice were monitored for systolic and diastolic blood pressure before surgery, postoperatively, and after administration using a tail-cuff non-invasive blood pressure analysis system (BP-2000 system, VisiTech Systems, Singapore) [38]. Blood pressure was measured three times under rest state and the average values were obtained [39]. 

### 4.6. Cardiac Function Assessment

After three weeks of continuous administration, all animals were fasted for 12 h. After isoflurane anesthesia, the interventricular septum (IVS), left ventricular posterior wall (LVPW), ejection fraction (EF), and fractional shortening (FS) were measured using a color ultrasonic Doppler diagnostic system (Mindray-M9Vet) [40,41].

### 4.7. Tissue and Blood Sample Collection

After fasting for 12 h, the mice were intraperitoneally anesthetized with 1% pentobarbital. Subsequently, blood samples were quickly drawn from the abdominal aorta. The heart, liver, spleen, lungs, and brain tissues were rapidly extracted. Some heart tissues were preserved in liquid nitrogen, whereas others were immobilized with 10% formaldehyde for further analysis [42]. 

### 4.8. Enrichment and Establishment of Drug–Targets–Disease Networks

The intersection targets of CA and RH were analyzed using Venny. A protein–protein interaction (PPI) network was built by using the STRING database. Common targets in the PPI network were visualized using Cytoscape. The pathways were identified using GO and KEGG analyses [14].

### 4.9. Molecular Docking 

The target structure of the anti-RH mechanism was downloaded from the Protein Data Bank (PDB). The binding capacity between CA and its targets was determined using the AutoDock 1.5.7 software. The results were visualized using PyMol [15]. The crystal structures of MMP9 (ID: 1L6J), TNF α (ID: 3L9J), COX2 (ID: 5IKR), IL-17 (ID: 7AMA), toll-like receptor 4 (TLR4) (ID: 2Z62), nitric oxide synthase 3 (NOS3) (ID: 1M9M), and the jun proto-oncogene, AP-1 transcription factor subunit (JUN) (ID: 5FV8), were acquired from the PDB database.

### 4.10. Histopathological Analysis

Heart, liver, and other tissues were cleaned with physiological saline, immobilized in 10% formaldehyde solution, and embedded in paraffin. Slices (3 μm) were cut and stained with hematoxylin and eosin [43]. Histomorphological changes in various tissues were observed by optical microscopy. 

### 4.11. ELLSA

Blood was centrifuged at 3000× *g*/5 min to obtain serum. The contents of TNF α and IL-17 were quantitated by an enzyme-linked immunosorbent assay (ELISA) kit on the basis of the manufacturer’s protocol [44].

### 4.12. Cell Culturing

H9C2 cells were purchased from the American Type Culture Collection (ATCC) and were cultured in Dulbecco’s modified Eagle’s Medium (DMEM) supplemented with 10% fetal bovine serum (FBS) and 1% penicillin and streptomycin (PS). The cells were maintained in a 37 °C thermostatic incubator with 5% CO_2_ [45].

### 4.13. CCK-8 Assay

The H9C2 cells were seeded in 96-well plates at a density of 3000 cells/well and incubated overnight [42]. Ang II and CA were added with different concentrations and incubated for 24 h. Cell viability was measured after adding a cell counting kit-8 (CCK-8) solution for 1.5 h.

### 4.14. Western Blot Analysis

RIPA solution containing protease inhibitors was used to lyse the heart tissue and H9C2 cells to obtain the total proteins. A BCA kit was used to determine protein concentrations. Subsequently, the protein sample was mixed with the loading buffer and boiled for denaturation. Proteins were isolated using sodium dodecyl sulfate–polyacrylamide gel electrophoresis (SDS-PAGE), and transferred to PVDF membranes. Then, the membranes were sealed in 5% skim milk at room temperature for 1 h. Primary antibodies against MMP9 (dilution at 1:1000), TNF α (dilution at 1:1000), COX2 (dilution at 1:1000), and IL-17 (dilution at 1:1000) were used to incubate the membranes at 4 °C overnight. After incubation with the corresponding secondary antibody at room temperature for 1 h, the protein bands were observed using enhanced chemiluminescence (ECL) reagents and analyzed using Image Lab 6.1 software [46,47].

### 4.15. Statistical Analysis

GraphPad Prism 8.0 analyzed all data, and all results are presented as mean ± standard deviation (SD). One-way ANOVA and the Mann–Whitney test were used to analyze significant differences. *p* < 0.05 represents statistical significance.

## 5. Conclusions

In this study, we explored the efficacy of CA in treating cardiac inflammation and hypertrophy caused by RH. Through the integrated application of network pharmacology and experimental validation, we elucidated that CA can effectively prevent RH-induced myocardial hypertrophy by suppressing inflammatory responses. Our study not only provides a theoretical basis for the CA treatment of myocardial hypertrophy caused by RH, but also provides a possibility for the discovery of novel drugs with the dual functions of reducing blood pressure and alleviating myocardial hypertrophy.

## Figures and Tables

**Figure 1 ijms-25-10063-f001:**
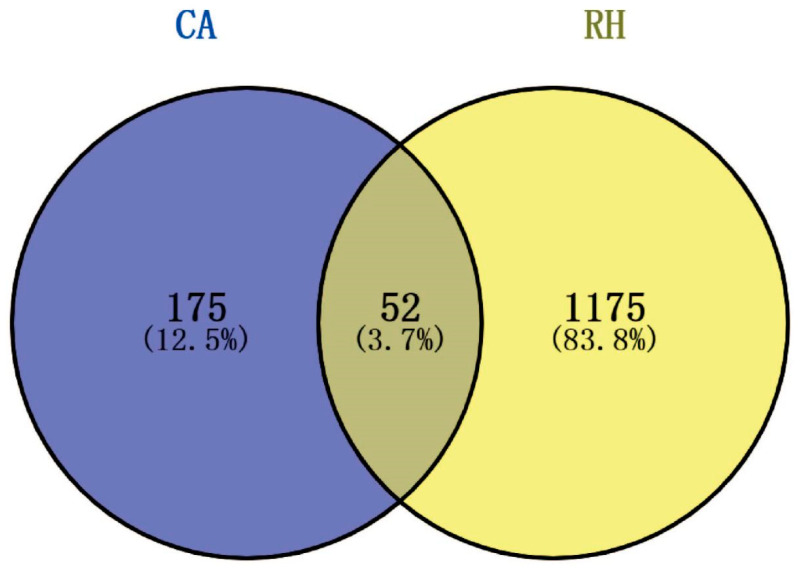
Venn diagram of intersection targets of CA and RH. Blue represents the targets of CA and yellow represents the targets of RH. There were 52 intersecting targets.

**Figure 2 ijms-25-10063-f002:**
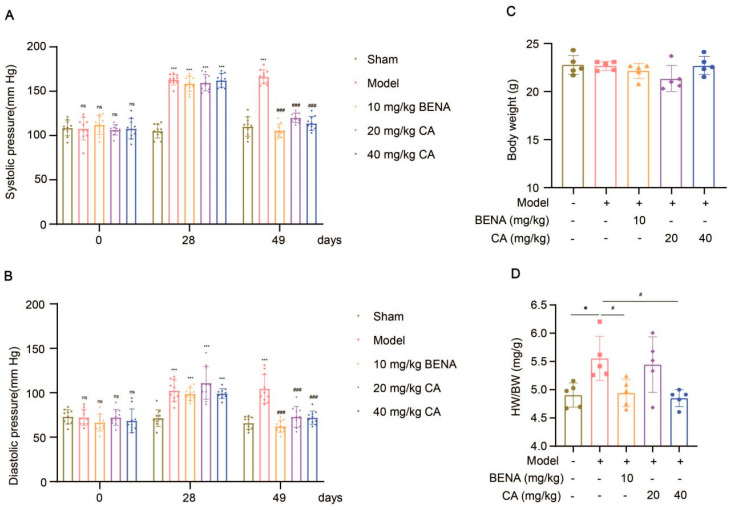
Effects of CA on cardiovascular parameters in RH mice. Systolic (**A**) and diastolic (**B**) blood pressures were monitored before surgery, four weeks after surgery, and three weeks after administration (*n* = 10). After fasting for 12 h, the body weight of each mice was measured (**C**), and their hearts were weighed to obtain the HW/BW ratio (**D**) (*n* = 5). In the quantitative chart, green circles represent sham group, pink squares represent model group, orange triangles represent 10 mg/kg BENA group, purple rhombuses represent 20 mg/kg CA group, and blue hexagons represent 40 mg/kg CA group. The values are mean ± SD. * *p* < 0.05, and *** *p* < 0.001 vs. sham; # *p* < 0.05, and ### *p* < 0.001 vs. model. ns = *p* > 0.05.

**Figure 3 ijms-25-10063-f003:**
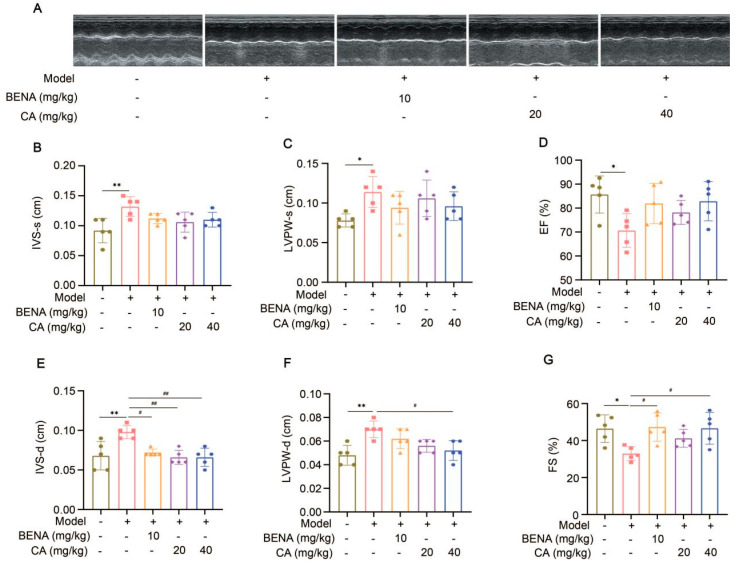
Effects of CA on echocardiographic indexes in RH mice. Representations of echocardiographic images are shown (**A**). From left to right, they are the sham group, model group, 10 mg/kg BENA group, and 20 mg/kg and 40 mg/kg CA groups, respectively. Statistics for IVS-s (**B**), LVPW-s (**C**), EF (**D**), IVS-d (**E**), LVPW-d (**F**), and FS (**G**). In the quantitative chart, green circles represent sham group, pink squares represent model group, orange triangles represent 10 mg/kg BENA group, purple rhombuses represent 20 mg/kg CA group, and blue hexagons represent 40 mg/kg CA group. The values are mean ± SD, *n* = 5. * *p* < 0.05, and ** *p* < 0.01 vs. sham; # *p* < 0.05, and ## *p* < 0.01 vs. model.

**Figure 4 ijms-25-10063-f004:**
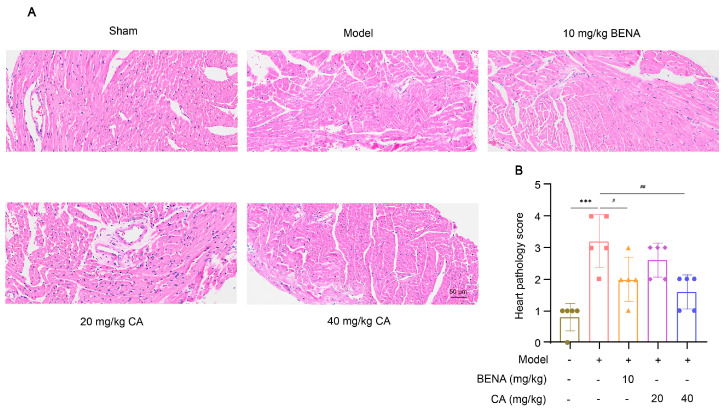
Effect of CA on cardiac histological changes in RH mice. H&E staining was used to observe the pathological changes in heart tissues in the sham group, model group, 10 mg/kg BENA group, and 20 mg/kg and 40 mg/kg CA groups. Representative images of hearts (**A**) are shown. The statistics of cardiac pathology scores (**B**) are also displayed. The scale bar represents 50 µm. In the quantitative chart, green circles represent sham group, pink squares represent model group, orange triangles represent 10 mg/kg BENA group, purple rhombuses represent 20 mg/kg CA group, and blue hexagons represent 40 mg/kg CA group. The values are mean ± SD, *n* = 5. *** *p* < 0.001 vs. sham; # *p* < 0.05, and ## *p* < 0.01 vs. model.

**Figure 5 ijms-25-10063-f005:**
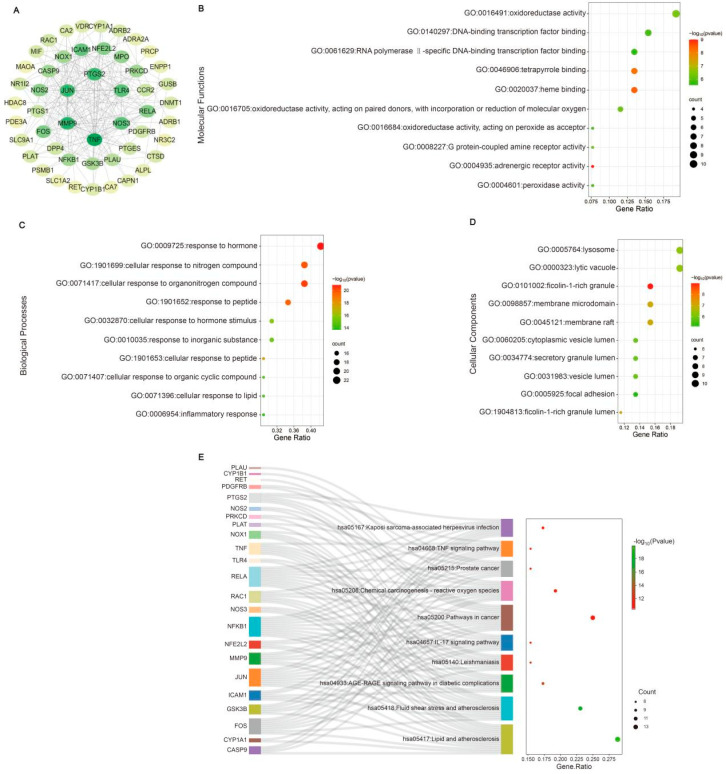
Network pharmacology analysis of CA in alleviating cardiac hypertrophy in mice with RH. A PPI network was established. The innermost nodes represent the CA–RH intersection targets with a degree of >20 (**A**). The top ten GO enrichment items of CA anti-RH core targets are reflected in MFs (**B**), BPs (**C**), and CCs (**D**). The top ten KEGG enrichment terms of core targets for CA against RH (**E**) are also shown.

**Figure 6 ijms-25-10063-f006:**
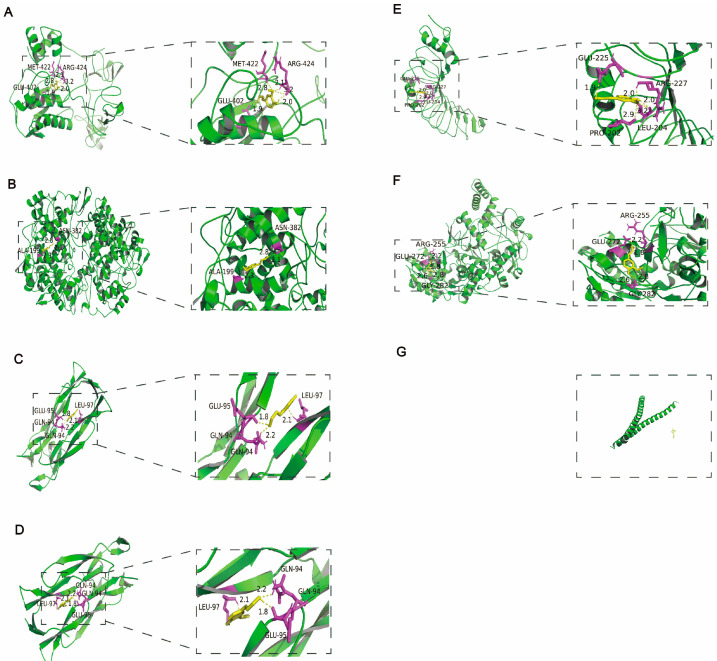
Molecular docking of CA with the core targets. Molecular docking of CA with the top six intersection targets. CA interacts with MMP9 via GLU-402, MET-422, and ARG-424 (**A**). CA interacts with COX2 via ALA-199, and ASN-382 (**B**). CA interacts with TNF α via GLU-94, GLU-95, and LEU-97 (**C**). CA interacts with IL-17 via GLU-94, GLU-95, and LEU-97 (**D**). CA interacts with TLR4 via GLU-225, ARG-227, PRO-202, and LEU-204 (**E**). CA interacts with NOS3 via GLU-272, ARG-255, and GLY-282 (**F**). CA and JUN have no interaction (**G**).

**Figure 7 ijms-25-10063-f007:**
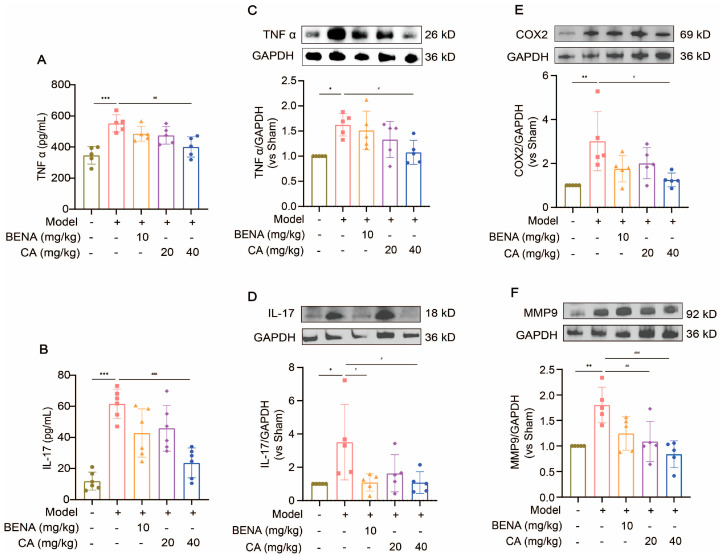
CA inhibited the expressions of TNF and IL-17 signaling pathway proteins. The serum levels of TNF α (**A**) and IL-17 (**B**) were measured using commercially available ELLSA kits after the administration of 10 mg/kg of BENA, 20 mg/kg of CA, and 40 mg/kg of CA, respectively, to the RH mice. The expressions of TNF α (**C**), IL-17 (**D**), COX2 (**E**), and MMP9 (**F**) were detected by Western blot analysis after the administration of 10 mg/kg of BENA, 20 mg/kg of CA, and 40 mg/kg of CA, respectively, to the RH mice. The quantification of normalized TNF α, IL-17, COX2, and MMP9 was performed. In the quantitative chart, green circles represent sham group, pink squares represent model group, orange triangles represent 10 mg/kg BENA group, purple rhombuses represent 20 mg/kg CA group, and blue hexagons represent 40 mg/kg CA group. The values are mean ± SD, *n* = 5. * *p* < 0.05, ** *p* < 0.01, and *** *p* < 0.001 vs. sham; # *p* < 0.05, ## *p* < 0.01, and ### *p* < 0.001 vs. model.

**Figure 8 ijms-25-10063-f008:**
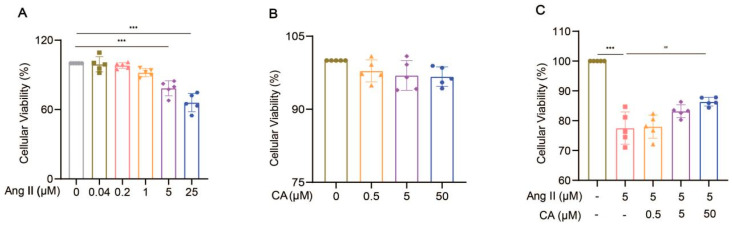
Effect of CA on the viability of H9C2 cells induced Ang II. The cell viability of the H9C2 cells treated with Ang II (0.04, 0.2, 1, 5, and 25 μM) (**A**) and CA (0.5, 5, and 50 μM) (**B**) was analyzed by CCK8. The H9C2 cells were treated with CA (0.5, 5, and 50 μM) in the presence of 5 μM Ang II (**C**). The cell viability of the 5 μM Ang II-induced H9C2 cells treated with CA (0.5, 5, and 50 μM) was detected by CCK8 assay. In quantitative (**A**), gray circles represent control, green squares represent 0.04 μM Ang II, pink upright triangles represent 0.02 μM Ang II, orange inverted triangles represent 1 μM Ang II, purple rhombuses represent 5 μM Ang II, and blue hexagons represent 25 μM Ang II. In quantitative (**B**), green circles represent control, orange triangles represent 0.5 μM CA, purple rhombuses represent 5 μM CA, and blue hexagons represent 50 μM CA. In quantitative (**C**), green circles represent control, pink squares represent 5 μM Ang II, orange triangles represent 5 μM Ang II + 0.5 μM CA, purple rhombuses represent 5 μM Ang II + 5 μM CA, and blue hexagons represent 5 μM Ang II + 50 μM CA. The values are mean ± SD, *n* = 5. *** *p* < 0.001 vs. control; ## *p* < 0.01 vs. 5 μM Ang II.

**Figure 9 ijms-25-10063-f009:**
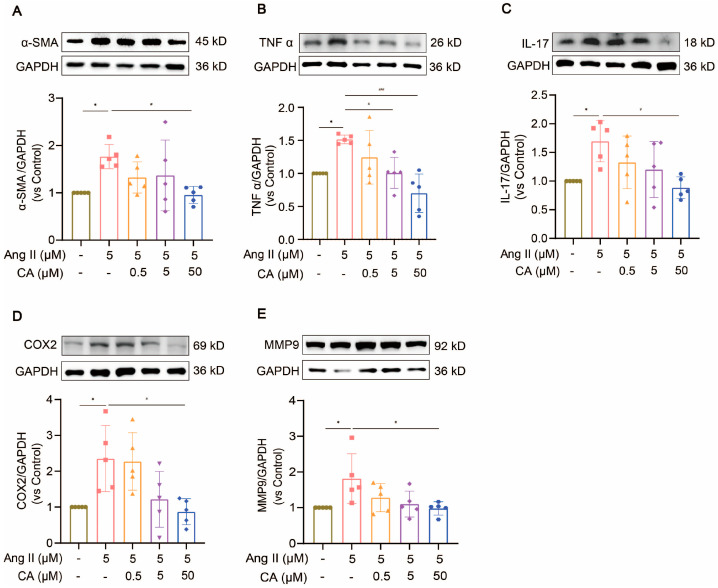
The inhibitory effect of CA on Ang II-stimulated hypertrophy of H9C2 cells. H9C2 cells were exposed to different concentrations of CA (0.5, 5, and 50 μM) under the stimulation of 5 μM Ang II. The expressions of α-SMA (**A**), TNF α (**B**), IL-17 (**C**), COX2 (**D**), and MMP9 (**E**) were detected by Western blot analysis. The quantification of normalized α-SMA, TNF α, IL-17, COX2, and MMP9 was performed. In the quantitative chart, green circles represent control, pink squares represent 5 μM Ang II, orange triangles represent 5 μM Ang II + 0.5 μM CA, purple rhombuses represent 5 μM Ang II + 5 μM CA, and blue hexagons represent 5 μM Ang II + 50 μM CA The values are mean ± SD, *n* = 5. * *p* < 0.05 vs. control; # *p* < 0.05, and ### *p* < 0.001 vs. 5 μM Ang II.

**Figure 10 ijms-25-10063-f010:**
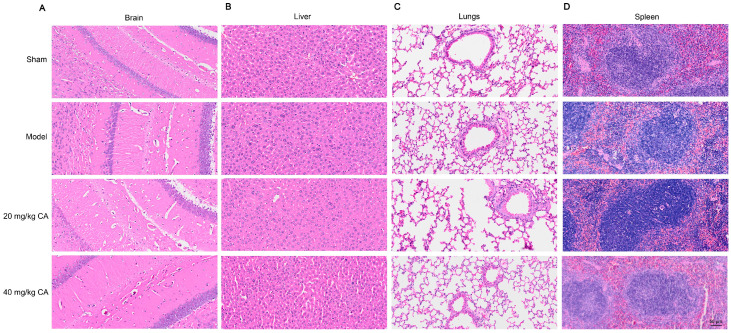
H&E staining of brain, liver, lungs, and spleen after CA treatment. Pathological changes in brain, liver, lungs, and spleen tissues from the sham group, model group, and 20 mg/kg and 40 mg/kg CA groups were evaluated by H&E staining. Representative images of the brain (**A**), liver (**B**), lungs (**C**), and spleen (**D**) are shown. The scale bar represents 50 µm.

**Table 1 ijms-25-10063-t001:** Intersection targets of CA and RH.

Target	Name
ADRA2A	Adrenoceptor alpha 2A
ADRB1	Adrenoceptor beta 1
ADRB2	Adrenoceptor beta 2
MAOA	Monoamine oxidase A
PTGS1	Prostaglandin-endoperoxide synthase 1
PTGS2	Prostaglandin-endoperoxide synthase 2
CA2	Carbonic anhydrase 2
ADRA1D	Adrenoceptor alpha 1D
DPP4	Dipeptidyl peptidase 4
NOS2	Nitric oxide synthase 2
NOS3	Nitric oxide synthase 3
RELA	RELA proto-oncogene, NF-KB subunit
MIF	Macrophage migration inhibitory factor
CA7	Carbonic anhydrase 7
ALPL	Alkaline phosphatase, biomineralization associated
ICAM1	Intercellular adhesion molecule 1
GSK3B	Glycogen synthase kinase 3 beta
CASP9	caspase 9
TNF	Tumor necrosis factor
RAC1	Rac family small GTPase 1
MMP9	Matrix metallopeptidase 9
PDE3A	Phosphodiesterase 3A
PRKCD	Protein kinase C delta
RET	Ret proto-oncogene
DNMT1	DNA methyltransferase 1
NFE2L2	NFE2 like BZIP transcription factor 2
NFKB1	Nuclear factor kappa B subunit 1
PSMB1	Proteasome 20S subunit beta 1
NR1I2	Nuclear receptor subfamily 1 group I member 2
GUSB	Glucuronidase beta
CTSD	Cathepsin D
ENPP1	Ectonucleotide pyrophosphatase/phosphodiesterase 1
NOX1	NADPH oxidase 1
PRCP	Prolylcarboxypeptidase
PDGFRB	Platelet derived growth factor receptor beta
HDAC8	Histone deacetylase 8
NR3C2	Nuclear receptor subfamily 3 group C member 2
SLC9A1	Solute carrier family 9 member A1
PLAU	Plasminogen activator, urokinase
PLAT	Plasminogen activator, tissue type
VDR	Vitamin D receptor
SLC1A2	Solute carrier family 1 member 2
CCR2	C-C motif chemokine receptor 2
CAPN1	Calpain 1
CACNA1H	Calcium voltage-gated channel subunit alpha1 H
FOS	Fos proto-oncogene, AP-1 transcription factor subunit
TLR4	Toll-like receptor 4
PTGES	Prostaglandin E synthase
JUN	Jun proto-oncogene, AP-1 transcription factor subunit
MPO	Myeloperoxidase
CYP1A1	Cytochrome P450 family 1 subfamily A member 1
CYP1B1	Cytochrome P450 family 1 subfamily B member 1

**Table 2 ijms-25-10063-t002:** Information on core targets.

Target	Name	Degree
TNF	Tumor necrosis factor	32
MMP9	Matrix metallopeptidase 9	26
PTGS2	Prostaglandin-endoperoxide synthase 2	24
JUN	Jun proto-oncogene, AP-1 transcription factor subunit	24
TLR4	Toll-like receptor 4	21
NOS3	Nitric oxide synthase 3	21

## Data Availability

Data will be provided upon request in the Appendix A.

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
