# Peer review of "Mechanism Actions of Coniferyl Alcohol in Improving Cardiac Dysfunction in Renovascular Hypertension Studied by Experimental Verification and Network Pharmacology"

_ijms, 2024, doi:10.3390/ijms251810063_

Round 1

Reviewer 1 Report

Comments and Suggestions for Authors

In the study the mechanisms involved in the effects of coniferyl alcohol (CA) in improving cardiac dysfunction in renovascular hypertension (RH) have been investigated.

I have some questions and comments:

1. Your experimental data indicate that CA improves cardiac inflammation and myocardial hypertrophy in RH by regulating the expression of MMP-9, COX-2, TNF α, and IL-17. In which way and by which molecular mechanisms regulates CA the expression of mentioned proteins?

2. Using molecular docking techniques you demonstrated excellent binding affinity and intermolecular interactions between CA and TNF, IL-17, COX-2, and MMP-9 as possible targets. In your opinion, do intermolecular interactions between CA and these proteins play a role in the mechanisms of action of CA?

3. To normalize the data obtained after the Western blot analysis for individual proteins, you used GAPDH (protein to GAPDH ratio). However, GAPDH levels in individual samples (especially in tissue samples) showed a high degree of variability and thus significantly influenced the evaluation and interpretation of data for a given specific protein.

Are the presented Western blot records representative? In the legend to the figure, you defined that number of samples used in each group was 5 (n=5). In such way you have several other records.

4. In results you wrote that TNF-alpha and IL-17 levels were determined in plasma. However, in methods you described the preparation of serum.

5. Original Western blot images show not very specific reactions of several used antibodies. Special example is antibody against IL-17. The source of this antibody is also not described in Materials and Methods.

Author Response

Comment 1: Your experimental data indicate that CA improves cardiac inflammation and myocardial hypertrophy in RH by regulating the expression of MMP-9, COX-2, TNF α, and IL-17. In which way and by which molecular mechanisms regulates CA the expression of mentioned proteins?

Authors’ reply: Thank the reviewer for this great comment. In the study of some small molecule drugs, it is common for drugs to affect the stability of proteins by binding with them or by affecting abundance and activity. Examples are as follows. Britannin specifically binds to ZEB1 and induces its ubiquitination to disturbe protein stability and promote ZEB1 protein degradation (PMID: 35839735). Propofol may antagonize the role of lipopolysaccharide (LPS) in activating HIF-1α through attenuating the protein stability of HIF-1α (PMID: 28426124). Castor zinc finger 1 (CASZ1) exerts its tumor-suppressive effect by directly interacting with RAF1 and reducing the protein stability of RAF1(PMID: 29506567). Therefore, CA, as a small molecular drug derived from traditional Chinese medicine, may have a similar pattern of action to the above-mentioned drugs. CA may regulate protein expression by binding to MMP9, COX2, TNF α and IL-17, inducing ubiquitination, phosphorylation, acetylation or reducing their abundance, thereby affecting protein stability. We have added this part to the discussion section so that readers can better understand the molecular mechanisms by which CA regulates the expression of MMP9, COX2, TNF α and IL-17. Add as follows:

Previous studies of small molecule drugs in diseases have shown that drugs often affect protein stability by binding to proteins and affecting abundance, which are more common in inflammatory, tumor, and autoimmune diseases. It is common for drugs to regulate protein expression by interacting with proteins to induce ubiquitination, phosphorylation and acetylation, or to reduce protein abundance to affect stability. Therefore, CA, as a small molecule drug derived from traditional Chinese medicine, may have a similar pattern of action with other small molecules. In cardiac inflammation caused by renovascular hypertension, CA may interact with TNF α, IL-17, COX2 and MMP9 to affect their protein stability and reduce their expression, thus playing a role in improving inflammation. The interaction between CA and TNF α, IL-17, COX2 and MMP9 needs to be further explored through pull-down, surface plasmon resonance analysis and other experiments, which will be the primary problem to be solved in our subsequent research (4. Discussion, paragraph 4, line 392-404).

Comment 2: Using molecular docking techniques you demonstrated excellent binding affinity and intermolecular interactions between CA and TNF, IL-17, COX-2, and MMP-9 as possible targets. In your opinion, do intermolecular interactions between CA and these proteins play a role in the mechanisms of action of CA?

Authors’ reply: Thank the reviewer for this good comment. Molecular docking results showed that CA could interact with IL-17, TNF, MMP9 and COX2. The experimental verification also proved that CA did reduce the expression of IL-17, TNF, MMP9 and COX2. As we mentioned in the previous question, other small molecule drugs play a role in improving inflammation in inflammatory diseases by binding to proteins or reducing their abundance to interfere with protein stability, thereby regulating its expression. Therefore, in cardiac inflammation caused by renovascular hypertension, CA may interact with TNF, IL-17, COX-2 and MMP-9 to affect their protein stability and reduce their expression, thus playing a role in improving inflammation. The interaction between CA and TNF α, IL-17, COX2 and MMP9 needs to be further explored through pull-down, surface plasmon resonance analysis and other experiments, which will be the primary problem to be solved in our subsequent research.

Comment 3:  To normalize the data obtained after the Western blot analysis for individual proteins, you used GAPDH (protein to GAPDH ratio). However, GAPDH levels in individual samples (especially in tissue samples) showed a high degree of variability and thus significantly influenced the evaluation and interpretation of data for a given specific protein.

Are the presented Western blot records representative? In the legend to the figure, you defined that number of samples used in each group was 5 (n=5). In such way you have several other records.

Authors’ reply: Thank the reviewer for this valuable comment. The explanation for selecting GAPDH as the internal control is as follows. GAPDH is an enzyme that plays a key role in cellular metabolism glycolysis, and its expression is usually high, and the expression level in the same cell or tissue is generally constant, and rarely affected by external inducers. Its expression may change only under certain conditions, such as hypoxia and the presence of diseases such as diabetes, so it is not recommended as an internal control in these cases. Renovascular hypertension is an increase in blood pressure caused by renal artery stenosis, pressure overload, resulting in dysfunction of heart and kidneys, with minimal relationship to hypoxia and glycolysis. Therefore, GAPDH can be selected as the internal control in the model of renovascular hypertension. Moreover, western blot analysis of heart tissues in the literature on cardiac inflammation and myocardial hypertrophy also extensively used GAPDH as an internal control (quote: PMID: 31518162, PMID: 10406830, PMID: 33251220, PMID: 34371252, PMID: 37116729, PMID: 37182595).

We have replaced the representative blots of COX2, IL-17 and MMP9 in Figure 7, and provided their original images in the supplementary material (3. Results, Figure 7, line 296).

The corrected Figure 7 is shown below.

Comment 4:  In results you wrote that TNF-alpha and IL-17 levels were determined in plasma. However, in methods you described the preparation of serum.

Authors’ reply: Thank the reviewer for this kind reminder and we are sorry for this mistake. We have corrected the plasma levels of TNF α and IL-17 to the serum levels of TNF α and IL-17 in the results (3. Results, line 289).  

Comment 5:  Original Western blot images show not very specific reactions of several used antibodies. Special example is antibody against IL-17. The source of this antibody is also not described in Materials and Methods.

Authors’ reply: Thank the reviewer for this kind reminder. IL-17 and TNF α are pro-inflammatory factors, which are secreted by cells and participate in the inflammatory process. Therefore, IL-17 and TNF α are mostly detected by ELLSA method. Western blot analysis of IL-17 and TNF α in the heart tissues of mice from different individuals is difficult to some extent, but the results of western blot analysis of IL-17 and TNF α in this study were consistent with those of ELLSA. In addition, the antibodies used in this study are also non-monoclonal antibodies, so there are non-specific bands. Meanwhile, we have supplemented the description of IL-17 antibody sources in the materials and methods (2. Materials and methods, 2.1. Reagents and materials, line 92). IL-17 antibody information is as follows:

IL-17 antibody (DF6127) was got from Affinity Biosciences (OH, United States).

Reviewer 2 Report

Comments and Suggestions for Authors

1.The Authors are encouraged to discuss better the principles and the practical applications of Network Pharmacology in cardiovascular clinical setting and how their findings sum to the already available data to improve the novelty and relevance of the manuscript (quote: PMID: 34895945, PMID: 30867426, PMID: 33785068, PMID: 32307915)

2.Legends of Figure 5 are difficult to read. I suggest to increase the fold.

3. The Authors are encouraged to add detailed figure legends for each figure and related panels.

4. Discussion should start with 2-3 sentences which highlights the main findings. The Authors are encouraged to remove redundant background information.

Comments on the Quality of English Language

Minor editing is required

Author Response

Comment 1: The Authors are encouraged to discuss better the principles and the practical applications of Network Pharmacology in cardiovascular clinical setting and how their findings sum to the already available data to improve the novelty and relevance of the manuscript (quote: PMID: 34895945, PMID: 30867426, PMID: 33785068, PMID: 32307915)

Authors’ reply: Thank the reviewer for this valuable comment.We have expanded the principles and practical applications of network pharmacology in cardiovascular clinical settings. The additions are as follows:

Network pharmacology is an innovation in the research way of "diseases-targets-drugs" in medicine. It integrates and summarizes information from existing databases through a series of methods to determine effective drug intervention treatment for diseases. The multi-target pattern of network pharmacology provides a more reasonable explanation for diseases and complications, facilitating disease identification and drug combination therapy. Network pharmacology has been widely applied in cardiovascular diseases, such as identifying the effective components and potential pharmacological mechanisms of Danshen Decoction in the treatment of cardiovascular diseases, clarifying the effect of QiShenYiQi Dripping Pills (T101) in the treatment of heart failure depend on multi-components, multi-targets and multi-pathways, and revealing the efficacy and potential mechanism of anxia Baizhu Tianma Decoction in the treatment of hypertension (1. Introduction, paragraph 3, line 63-75) (quote: PMID: 34895945, PMID: 30867426, PMID: 33785068, PMID: 36332387, PMID: 36353495, PMID: 36534045).

Comment 2: Legends of Figure 5 are difficult to read. I suggest to increase the fold.

Authors’ reply: Thank the reviewer for this positive suggestion and we are sorry for this unclear expression. We have expanded the legends of Figure 5 and corrected it in the revised manuscript. The extensions are as follows:

Establish a PPI network. The innermost nodes represent CA-RH intersection targets with degree>20 (A). The top ten GO enrichment items of CA anti-RH core targets are reflected in MF (B), BP (C) and CC (D). Top ten KEGG enrichment terms of core targets for CA against RH (E) (line 270-273).

Comment 3: The Authors are encouraged to add detailed figure legends for each figure and related panels.

Authors’ reply: Thank the reviewer for this kind reminder. We have added detailed legends for each figure.

Figure 1. Venn diagram of intersection targets of CA and RH. Blue represents the target of CA and yellow represents the target of RH. There were 52 intersecting targets (line 218-219).

Figure 2. Systolic (A) and diastolic (B) blood pressures were monitored before surgery, four weeks after surgery, and three weeks after administration (n=10). After fasting for 12h, the body weight of each mice was measured (C), and the hearts were weighed to obtain the HW/BW ratio (D) (n=5). In the quantitative chart, green represents Sham group, pink represents Model group, yellow represents 10 mg/kg BENA group, purple represents 20 mg/kg CA group, and blue represents 40 mg/kg CA group. Values are mean ± SD. * p < 0.05, ** p < 0.01, *** p < 0.001 vs. Sham; # p < 0.05, ## p < 0.01, ### p < 0.001 vs. Model. ns= p > 0.05 (line 224-231).

Figure 3. Representation of echocardiographic images (A). From left to right, they were Sham group, Model group, 10 mg/kg BENA group, 20 mg/kg and 40 mg/kg CA group, respectively. Statistics for IVS-s (B), LVPW-s (C), EF (D), IVS-d (E), LVPW-d (F) and FS (G). In the quantitative chart, green represents Sham group, pink represents Model group, yellow represents 10 mg/kg BENA group, purple represents 20 mg/kg CA group, and blue represents 40 mg/kg CA group. Values are mean ± SD, n = 5. * p < 0.05, ** p < 0.01, *** p < 0.001 vs. Sham; # p < 0.05, ## p < 0.01, ### p < 0.001 vs. Model (line 234-240).

Figure 4. H&E staining was used to observe the pathological changes of heart tissues in Sham group, Model group, 10 mg/kg BENA group, 20 mg/kg and 40 mg/kg CA group. Representative images of hearts (A). Statistics of cardiac pathology scores (B). Scale bar represents 50 µm. In the quantitative chart, green represents Sham group, pink represents Model group, yellow represents 10 mg/kg BENA group, purple represents 20 mg/kg CA group, and blue represents 40 mg/kg CA group. Values are mean ± SD, n = 5. * p < 0.05, ** p < 0.01, *** p < 0.001 vs. Sham; # p < 0.05, ## p < 0.01, ### p < 0.001 vs. Model (line 243-249).

Figure 5. Establish a PPI network. The innermost nodes represent CA-RH intersection targets with degree>20 (A). The top ten GO enrichment items of CA anti-RH core targets are reflected in MF (B), BP (C) and CC (D). Top ten KEGG enrichment terms of core targets for CA against RH (E) (line 270-273).

Figure 6. Molecular docking of CA with the top six intersection targets. CA interacts with MMP9 via GLU-402, MET-422, and ARG-424 (A). CA interacts with COX2 via ALA-199, and ASN-382 (B). CA interacts with TNF α via GLU-94, GLU-95, and LEU-97 (C). CA interacts with IL-17 via GLU-94, GLU-95, and LEU-97 (D). CA interacts with TLR4 via GLU-225, ARG-227, PRO-202, and LEU-204 (E). CA interacts with NOS3 via GLU-272, ARG-255, and GLY-282 (F). CA and JUN have no interaction (G) (line 278-283).

Figure 7. The serum levels of TNF α (A) and IL-17 (B) were measured using the commercially available ELLSA kits after administration of RH mice at 10 mg/kg BENA, 20 mg/kg and 40 mg/kg CA, respectively. The expressions of TNF α (C), IL-17 (D), COX2 (E), and MMP9 (F) were detected by western blot analysis after administration of RH mice at 10 mg/kg BENA, 20 mg/kg and 40 mg/kg CA, respectively. Quantification of normalized TNF α, IL-17, COX2, and MMP9. In the quantitative chart, green represents Sham group, pink represents Model group, yellow represents 10 mg/kg BENA group, purple represents 20 mg/kg CA group, and blue represents 40 mg/kg CA group. Values are mean ± SD, n = 5. * p < 0.05, ** p < 0.01, *** p < 0.001 vs. Sham; # p < 0.05, ## p < 0.01, ### p < 0.001 vs. Model (line 298-307).

Figure 8. The cell viability of H9C2 cells treated with Ang II (0.04, 0.2, 1, 5, 25 μM) (A) and CA(0.5, 5, 50 μM) (B) was analyzed by CCK8. H9C2 cells were treated with CA(0.5, 5, 50 μM) in the presence of 5 μM Ang II (C). The cell viability of 5 μM Ang II-induced H9C2 cells treated with CA(0.5, 5, 50 μM) was detected by CCK8 assay. Values are mean ± SD, n = 5. * p < 0.05, ** p < 0.01, *** p < 0.001 vs. Control; # p < 0.05, ## p < 0.01, ### p < 0.001 vs. Ang II (line 322-327).

Figure 9. H9C2 cells were exposed to different concentrations of CA (0.5, 5, 50 μM) under the stimulation of 5 μM Ang II. The expressions of α-SMA (A), TNF α (B), IL-17 (C), COX2 (D), and MMP9 (E) were detected by western blot analysis. Quantification of normalized α-SMA, TNF α, IL-17, COX2, and MMP9. In the quantitative chart, green represents Control group, pink represents 5 μM Ang II group, yellow represents 0.5 μM CA group, purple represents 5 μM CA group, and blue represents 50 μM CA group. Values are mean ± SD, n = 5. * p < 0.05, ** p < 0.01, *** p < 0.001 vs. Control; # p < 0.05, ## p < 0.01, ### p < 0.001 vs. Ang II (line 330-337).

Figure 10. The pathological changes of brain, liver, lungs and spleen tissues in Sham group, Model group, 20 mg/kg and 40 mg/kg CA group were evaluated by H&E staining. Representative images of the brain (A), liver (B), lungs (C) and spleen (D). Scale bar represents 50 µm (line 345-348).

Comment 4: Discussion should start with 2-3 sentences which highlights the main findings. The Authors are encouraged to remove redundant background information.

Authors’ reply: Thank the reviewer for this valuable comment. We have revised the first paragraph of the discussion to read as follows:

In this study, based on reports of CA vasodilation, we demonstrated, for the first time, CA had the effect of lowering blood pressure and alleviating cardiac inflammation under RH model through the combination of network pharmacological analysis and experimental testing. We found that CA ameliorated cardiac inflammation and myocardial hypertrophy after RH by regulating the expressions of IL-17, TNF α, MMP9 and COX2 in both IL-17 and TNF signaling pathways. Moreover, similar verification was further obtained on H9C2 cells in vitro (4. Discussion, paragraph 1, line 350-356).

Round 2

Reviewer 1 Report

Comments and Suggestions for Authors

Authors answered my questions and comments and included some corresponding changes to the revised manuscript.